# Chitooligosaccharides Modulate Glucose-Lipid Metabolism by Suppressing SMYD3 Pathways and Regulating Gut Microflora

**DOI:** 10.3390/md18010069

**Published:** 2020-01-20

**Authors:** Qiutong Wang, Yajie Jiang, Xuegang Luo, Chang Wang, Nan Wang, Hongpeng He, Tongcun Zhang, Liehuan Chen

**Affiliations:** 1Key Lab of Industrial Fermentation Microbiology of the Ministry of Education & Tianjin Key Lab of Industrial Microbiology, College of Biotechnology, Tianjin University of Science and Technology, Tianjin 300457, China; wangqiutong1991@163.com (Q.W.); 13820754972@163.com (Y.J.); 18322595834@163.com (C.W.); wn929@tust.edu.cn (N.W.); hehongpeng@tust.edu.cn (H.H.); tony@tust.edu.cn (T.Z.); 2Tianjin Engineering Research Center of Microbial Metabolism and Fermentation Process Control, Tianjin 300457, China; 3College of Animal Sciences and Technology, Zhongkai Agricultural Engineering College, Guangzhou 510225, China; 4Guangzhou Youlan Marine Biological Technology Co., Ltd., Guangzhou 510530, China

**Keywords:** HMGCR, chitooligosaccharides, glycolipid metabolism disorder, intestinal microflora, SMYD3

## Abstract

Chitooligosaccharides (COS) have a variety of biological activities due to their positively charged amino groups. Studies have shown that COS have antidiabetic effects, but their molecular mechanism has not been fully elucidated. The present study confirmed that COS can reduce hyperglycemia and hyperlipidemia, prevent obesity, and enhance histological changes in the livers of mice with type 2 diabetes mellitus (T2DM). Additionally, treatment with COS can modulate the composition of the gut microbiota in the colon by altering the abundance of *Firmicutes*, *Bacteroidetes*, and *Proteobacteria*. Furthermore, in T2DM mice, treatment with COS can upregulate the cholesterol-degrading enzymes cholesterol 7-alpha-hydroxylase (CYP7A1) and incretin glucagon-like peptide 1 (GLP-1) while specifically inhibiting the transcription and expression of 3-hydroxy-3-methylglutaryl coenzyme A reductase (HMGCR), the key enzyme in cholesterol synthesis. Furthermore, using an oleic acid-induced hepatocyte steatosis model, we found that HMGCR can be directly transactivated by SET and MYND domain containing 3 (SMYD3), a transcriptional regulator, via 5′-CCCTCC-3′ element in the promoter. Overexpression of SMYD3 can suppress the inhibitory effect of COS on HMGCR, and COS might regulate HMGCR by inhibiting SMYD3, thereby exerting hypolipidemic functions. To the best of our knowledge, this study is the first to illustrate that COS mediate glucose and lipid metabolism disorders by regulating gut microbiota and SMYD3-mediated signaling pathways.

## 1. Introduction

With the accelerated pace of modern life and changes in dietary structure, abnormal glucose and lipid metabolism and the resulting metabolic diseases are increasingly jeopardizing human health [1]. Disorders of glucose and lipid metabolism caused by dyslipidemia and hyperglycemia promote the development of metabolic-related diseases, and they often affect each other [2]. Type 2 diabetes mellitus (T2DM) is a metabolic disorders characterized by hyperglycemia, insulin resistance, decreased glucose tolerance, and hyperlipidemia [3]. In addition, type 2 diabetes can cause serious complications, such as neuropathy, nephropathy, and cardiovascular and cerebrovascular diseases. Currently, the type 2 diabetes is mainly treated by oral hypoglycemic agents combined with diet and exercise. In addition, some functional oligosaccharides (such as inulin, arabinose, acarbose) have been found to be effective in the treatment of type 2 diabetes by regulating intestinal bacteria and glycolipid metabolism.

Chitooligosaccharides (COS) are oligomers of *N*-acetylglucosamine (GlcNAc) and glucosamine (GlcN) linked by β-1,4 glycosidic bonds with a degree of polymerization of 2–10: oligosaccharide β-(1,4)-2-amino-2-deoxy-d-glucose [4]. COS are formed by acid hydrolysis and the deacetylation of chitosan. It has a molecular weight of less than 2000 Da, is nontoxic, water-soluble, and easily absorbed by animals, and has good biocompatibility and biodegradability, a relatively small molecular mass, and stronger biological activity than chitosan [5,6,7]. Because COS are positively charged, they not only have good biocompatibility, anti-bacterial properties, and biodegradability but also have bio-adhesiveness and desorption capacity [8]. A large number of reports indicate that COS have potential applications in foods [9], pharmaceuticals [10], agriculture, and the environmental industries [11].

COS are the only positively charged oligosaccharides in nature up to now. Previous study has found that COS can bind fats through a positively charged basic amino group and, therefore, reduce cholesterol absorption and fat accumulation [12,13,14]. The effects of COS on reducing cholesterol may be related to the following two pathways: One is by upregulating cholesterol 7-alpha-hydroxylase (CYP7A1), liver X receptor a (LXRα), and peroxisome proliferator-activated receptor-a (PPARα) to promote the conversion of cholesterol to bile acids; the other is by downregulating the genes of 3-hydroxy-3-methylglutaryl coenzyme A reductase (HMGCR) to reduce the synthesis of cholesterol [15]. COS are often used to treat blood lipid abnormalities, regulate blood lipid balance, and protect the liver. Previous studies have found that COS can inhibit the secretions from pancreatic glands, enhance the activity of lipases, and increase the excretion of fat in feces by combining fats with bile acid cations.

Gut microflora represents a symbiotic ecosystem consisting of a host and a large number of intestinal microbes. With the in-depth development of high-throughput sequencing technology, people’s understanding of gut microflora is more comprehensive. Recent studies have shown that intestinal flora disorders can lead to the development of chronic metabolic diseases such as obesity and diabetes [16,17,18]. As the largest micro-ecological system in the human body, the gut microflora and its relationship with chronic diseases, such as obesity induced by a high-fat diet, have also become research hotspots in recent years.

SET and MYND domain containing 3 (SMYD3) is a histone methyltransferase discovered in 2004 that specifically dimethylates or trimethylates histone H3-lysine 4 [19]. Studies have shown that SMYD3 plays an important role in the development of tumors [20,21,22,23,24] and is highly expressed in many cancer cells but not in the corresponding normal tissues. Overexpression of SMYD3 promotes malignant transformation of cells and enhances their proliferation and migration. Specific inhibition of SMYD3 reduces cancer cell proliferation and migration of cancer cells and induces apoptosis [20,25]. In addition, previous studies found that SMYD3 not only plays an important role in tumors, such as liver cancer, but also participates in metabolism, the action of steroid hormones and the cardiovascular system [26,27,28], in which it is a key regulator [29].

However, thus far, the roles of gut microflora and SMYD3 in the hypoglycemic and lipid-lowering effects of COS are still not clear. In the present study, using mice with type 2 diabetes mellitus (T2DM) and a hepatocyte steatosis model, the effects of COS on gut microflora and the SMYD3-mediated transcriptional regulation of HMGCR were investigated. 

## 2. Results

### 2.1. COS Can Ameliorate Hyperglycemia and Dyslipidemia Induced by T2DM

To detect the hyperglycemia and hyperlipidemia effects of COS, T2DM mice were administered 140 mg/kg/d COS for 5 weeks, and a diverse variation in body weight was observed among the 3 groups. The T2DM group showed a significant increase in body weight gain, and the body weight gains of the COS groups were less than those of the T2DM group (Figure 1A).

As shown in Figure 1B, compared with those of the control group (NFD), the blood glucose levels in the T2DM group were significantly increased, while blood glucose levels in the mice treated with COS were significantly decreased compared with those of the T2DM group. In addition, compared with the NFD group, the T2DM mice had decreased insulin levels in serum, while the mice treated with COS had insulin levels that tended to be normal (Figure 1C).

T2DM is often accompanied by hyperlipidemia and that is characterized by serum total cholesterol (TC) ≥ 5.18 mmol/L, triglyceride (TG) ≥ 1.7 mmol/L, high-density lipoprotein cholesterol (HDL-C) < 1.04 mmol/L, and density lipoprotein cholesterol (LDL-C) ≥ 3.37 mmol/L, according to the Guidelines for the Prevention and Treatment of Abnormal Blood Lipid in Adults in China. Therefore, we also detected changes in serum TC, TG, LDL-C, and HDL-C levels. The results showed that the TC, TG, and LDL-C content levels in the T2DM group were significantly higher than those in the NFD group (*p* < 0.05 or *p* < 0.01). However, COS (140 mg/kg/d) treatment could significantly inhibit the elevation of serum TC, TG, and LDL-C levels (Figure 1D).

### 2.2. COS Has Potential Protection Effects on Liver and Renal Damages of Type 2 Diabetic Mice

As shown in Figure 2, the results of hematoxylin-eosin (HE) staining showed livers of mice in the NFD group had a well-organized structure, hepatic sinusoids were clearly visible, and hepatic cords were neatly arranged, whereas the structures of livers displayed damages in T2DM group and hepatocytes showed signs of necrosis. However, such hepatocyte steatosis was obviously alleviated by treating with COS (Figure 2A). In addition, the kidneys also changed compared with those of the NFD group of normal mice. The kidneys from the T2DM group mice mainly had increased glomerular capillary expansion and vacuole degeneration. Kidney inflammation was obviously alleviated by treating with COS compared with T2DM group. It could be concluded that COS has potential protection effects on liver and kidney injury induced by T2DM (Figure 2B).

### 2.3. COS Altered the T2DM-Induced Gut Microflora Dysbiosis

To detect whether COS affect gut microflora, changes in microbial community structure were analyzed. As shown in Figure 3, on the order lever, *Firmicutes*, *Bacteroidetes*, *Proteobacteria*, and *Verrucomicrobiales* occupy dominant positions in the intestine. Compared with the mice in the T2DM group, mice treated with COS had an increased the ratio of *Firmicutes* to *Bacteroidetes* in the intestine, an increased relative abundance of *Verrucomicrobiales,* and decreased abundance of endotoxin-bearing *Proteobacteria.*

### 2.4. COS Regulated Glucose-Lipid Metabolism in Liver and Renal Pathology

T2DM is always accompanied by glucose-lipid metabolism disorders. HMGCR is the rate-limiting enzyme in the synthesis of cholesterol, while CYP7A1 is an important enzyme involved in the cholesterol catabolism. Glucagon-like peptide-1 (GLP-1) is a brain–gut peptide secreted by ileal endocrine cells and plays an important role in the development of type 2 diabetes. SMYD3 is a transcriptional regulator participates in metabolism, action of steroid hormones and the cardiovascular system. To determine whether COS mediate lipid metabolism, the changes of both messenger RNA (mRNA) and protein levels of HMGCR, CYP7A1, SMYD3, and GLP-1 were detected. As shown in Figure 4, treatment with COS could upregulate the expression of CYP7A1 and GLP-1 in the T2DM mice and downregulate HMGCR and SMYD3 expression simultaneously.

### 2.5. COS-Regulated Lipid Metabolism in the HepG2 Steatosis Model

To evaluate the lipid-reducing effects of COS, an oleic acid-induced high steatosis model of HepG2 cells was applied in this research. As shown in Figure 5A, the Oil red staining showed that the oleic acid treatment (HF) caused severe fatty degeneration of HepG2 cells compared to the control group. After treatment with COS (COS+HF), high-fat cells had significantly reduced fat content.

To investigate the mechanism of the inhibitory action of COS in hepatic steatosis, HMGCR and SMYD3 were examined at the mRNA and protein expression levels. As shown in Figure 5B,C, the expression of HMGCR and SMYD3 increased in the high-fat cell model of the HF group, while treatment with COS could significantly reduce the expression of these two genes.

To understand the regulatory mechanisms by which the SMYD3 gene affects the cholesterol synthesis rate-limiting enzyme HMGCR, we silenced the expression of SMYD3 using small interfering RNA (siRNA) and detected the expression of HMGCR at the mRNA and protein levels by real-time PCR and Western blotting. The results showed that the expression of HMGCR decreased significantly after SMYD3 interference (Figure 5E–G). Conversely, the expression of HMGCR increased when SMYD3 was overexpressed, and treatment with COS significantly reduced the expression of HMGCR (Figure 5I,J).

Furthermore, the results of the luciferase reporter assay further showed that oleic acid could stimulate the transcriptional activity of HMGCR promoter, while COS could markedly inhibit the activity of the HMGCR promoter. In addition, overexpression of SMYD3 could enhance the transcriptional activity of HMGCR promoter and inhibit the effect of COS (Figure 5D,H).

### 2.6. SMYD3 Activates HMGCR Transcription via SBE Elements in the Promoter

By bioinformatics analysis, we found the presence of the SMYD3 binding element (SBE) CCCTCC sequence at position—404 of the HMGCR promoter (Figure 6A). To explore whether this binding site is a key component in the SMYD3-induced regulation of HMGCR promoter activity, we cloned the −1439 to +20 segment of the human HMGCR promoter and ligated it into the pGL3 plasmid, a widely used reporter in which the luciferase will be driven by inserted promoter. The results showed that, after SMYD3 silencing, HMGCR promoter activity significantly decreased (Figure 6B). Overexpressing SMYD3 significantly enhanced HMGCR promoter activity (Figure 6C).

To evaluate the importance of SBE (CCCTCC) in the HMGCR promoter, the SBE (CCCTCC) sequence was mutated as depicted in Figure 6A. The results from the analysis of the luciferase reporter after the promoter was mutated showed that overexpressing or silencing SMYD3 did not alter the activity of the mutant HMGCR promoter compared to that of the wild-type promoter (Figure 6B,C).

To further confirm the role of SMYD3 in the transcriptional regulation of HMGCR and whether SMYD3 directly interacts with the CCCTCC sequence in the HMGCR enhancer region, we used a chromatin immunoprecipitation (CHIP) experiment. We first designed and synthesized three fragments of the HMGCR enhancer region. The fragments from PCR of the CHIP products are described in Figure 6D. Fragment I was a control fragment that is 800 bp upstream of the transcription start site (TSS). Fragment II was the core promoter covering SBE (CCCTCC), and fragment III was the core promoter covering the TSS. As shown in Figure 6E, CHIP assays with antibodies against SMYD3 demonstrated that SMYD3 could bind both SBE and TSS region of HMGCR promoter, indicating that SMDY3 might be recruited together with RNA polymerase II (RNAPII) in the transactivation of HMGCR. Furthermore, the results from the quantitative analysis of CHIP products showed that the overexpressed SMYD3 bound to the fragment II region approximately 2.8-fold more than it did to fragment I (Figure 6F) and RNAPII was recruited approximately 2.5-fold more than that it did to fragment I (Figure 6G).

## 3. Discussion

Currently, the role of COS in lowering cholesterol has received increasing attention. Many studies have shown that abnormal blood glucose levels are accompanied by dyslipidemia-related disorders [30,31]. The lipid metabolism disorder was notable in the T2DM mice. In this study, treatment with COS could ameliorate blood sugar levels and blood lipid levels (Figure 1B,D).

GLP-1 is an incretin hormone secreted from intestinal L-cells, which stimulates glucose-dependent insulin [32] secretion, which inhibits glucagon secretion and enhances β cell growth [32,33,34]. Additionally, GLP-1 may improve insulin sensitivity in patients with type 2 diabetes and animal models [35]. CYP7A1 is the rate-limiting enzyme of the bile acid (BA) synthesis pathway, which is the pivotal mechanism for maintaining the balance between cholesterol and BA [36]. HMGCR is a key enzyme in the metabolism of glycolipids in the body and has been recognized as an important target for hypolipidemic drugs (statins). Recent studies have found that HMGCR-inhibiting statins have anti-pancreatic cancer effects [37]. It is noteworthy that HMGCR is also closely related to tumorigenesis. Interestingly, research has shown that SMYD3 not only plays an important role in tumors, such as liver cancer, but also acts as a key regulator of the cardiovascular system [29]. A study of abnormalities in tumor cell metabolism revealed that increased lipid production is a notable metabolic feature of rapidly proliferating tumor cells. Cholesterol is a key component of the cell membrane, and its excessive intake or synthesis plays an important role in the occurrence and development of tumors. What is more, statin anticholesterol drugs have also been confirmed by more and more literature as potential antitumor drugs [37,38]. Through bioinformatics, we found that HMGCR promoter has a binding site for SMYD3; therefore, we speculate that SMYD3 and HMGCR have regulatory relationships. The animal studies showed that, compared with the T2DM group, the group treated with COS showed inhibited transcriptional regulation of HMGCR and reduced SMYD3 expression, while treatment with COS upregulated the cholesterol-degrading enzymes CYP7A1 and incretin GLP-1 (Figure 4). The hepatocyte steatosis model studies showed that knocking down SMYD3 led to significantly decreased HMGCR promoter activity. However, overexpression of SMYD3 suppressed the inhibitory effect of COS on HMGCR (Figure 5B–H). These results revealed that HMGCR is directly transactivated by SMYD3 via 5′-CCCTCC-3′ elements on the promoter and that SMYD3 is likely an important regulator of glycolipid metabolism. COS play a cholesterol-lowering effect by inhibiting the transcriptional expression of HMGCR and SMYD3 and simultaneously activating CYP7A1.

The gut microflora is a complex microbial ecological group with important functions in immunity, anti-aging, antitumor, nutrient decomposition, and metabolism [39,40,41,42,43] that are important to maintain human health. Therefore, research on intestinal flora has received increasing attention. Functional oligosaccharides are mainly fermented by the intestinal flora in the distal digestive tract, and their function is closely related to the regulation of the intestinal flora. In this experiment, the results from the 16s rDNA sequencing showed that treatment with COS caused diversified changes in the colonic intestinal flora (Larsen et al. [17]). The proportion of *Mycobacterium sinensis* and *Clostridium*—like *bacilli* in the intestinal tract of T2DM patients was significantly reduced, and the proportion of *Bacteroides*/*Firmicutes* was correlated with blood glucose levels. At the order level, the number of fecal *Burkholderiales* in the T2DM mice increased significantly, whereas some beneficial bacteria, such as *Verrucomicrobiales* microbes, decreased significantly in the normal group. *Verrucomicrobiales* play an important role in the pathogenesis of obesity and type 2 diabetes, and it would be much higher in the gut of healthy mammals than in humans and mice with obesity or type 2 diabetes. In the T2DM model group, the number of *Verrucomicrobiales* in the intestines of the mice was significantly lower than it was in the mice of the NFD group, while the number of *Verrucomicrobiales* in the intestines of T2DM mice treated with COS was increased. The T2DM mice treated with COS had significantly reduced *Burkholderiales* content, which reduced their incidence of enteritis. Furthermore, the ratio of *Bacteroidetes* to *Firmicutes*, which is considered to be one of the important factors of obesity, was decreased in the T2DM model group, while treatment of the T2DM mice with COS improved this ratio (Figure 3).

In summary, this paper was the first to find that SMYD3 can regulate glycolipid metabolism enzymes such as HMGCR and plays an important role in diabetes. COS can exert their hypoglycemic and lipid-lowering effects on the intestinal flora and the SMYD3/HMGCR pathway. This study sheds light on T2DM prevention and treatment.

## 4. Materials and Methods

### 4.1. Cell Culture and Establishment of High-Fat Cell Model

HepG2 cells were cultured in Dulbecco’s modified Eagle’s medium (DMEM) (Gibco, GrandIsland, NY, USA) (Appendix A). Ten percent fetal bovine serum (Gibco, USA) and 100 U/mL penicillin G and 0.1 mg/mL streptomycin (Gibco, GrandIsland, NY, USA) were added to the DMEM. The cells were cultured in a humidified cell incubator containing 5% carbon dioxide at 37 °C. Hepatocytes were treated with 0.5 mM oleic acid for 24 h and then COS, and were incubated in cell culture plates for 24 h.

### 4.2. Cell Transfection

The plasmid pcDNA5-TO/TAP-DEST-SMYD3 was a gift from Professor Philip Tucker (Institute for Cellular and Molecular Biology, University of Texas, Austin, TX, USA). The siRNA was synthesized by the Ribo Biological Company (Guangzhou, China) using the following sequences: The siRNA targeting the SMYD3 (si-SMYD3), sense strand, 5′-CAAGGAUGCUGAUAUGCUAdTdT-3′, and the antisense strand, 3′-dTdT GUUCCUACGACUAUACGAU-5′. The sequence information of the control siRNA (si-control) was not provided by the manufacturer. Transient transfection was performed using TurboFect reagent (Thermo Fisher Scientific, Inc., Waltham, MA, USA) following the manufacturer’s instructions.

### 4.3. Animals and Treatment

Thirty male Kun Ming mice (4 weeks, about 18–20 g each, purchased from the Experimental Animal Center of Military Medical College, permission number: SCXK (military) 2007-004) were randomly divided into a control group (NFD), hyperglycemic group (T2DM), and chitooligosaccharides group (COS, 140 mg/kg/d). One week after the mice were given free access to nutrition and water, for adaptive feeding, the mice in the NFD group were fed a basic diet (crude protein (%) ≥ 18.0; lysine (%) ≥ 0.82; crude fat (%) ≥ 4.0; crude ash (%) ≤ 8.0; crude fiber (%) ≤ 5.0; moisture (%) ≤ 10.0; calcium (%) 1.0~1.8; common salt (%) 0.3~0.8; phosphorus (%) 0.6~1.2), while the T2DM model and COS group were fed a high-fat diet (basic diet, 78.8%; lard, 10%; egg yolk powder, 10%; cholesterol, 1%; and deoxycholate, 0.2%) during the experiment. The T2DM group and COS group mice were intraperitoneally injected with 100 mg/kg streptozotocin (STZ, dissolved in citrate buffer, pH 4.4) in the 2nd week to establish T2DM model, which was deemed successful when the mouse fasting blood glucose (FBG) levels were ≥11.1 mmol·L. Then, the NFD and T2DM group mice were intragastrically administered normal saline, and the COS group mice were intragastrically administered with snow crab-derived chitooligosaccharides (Guangzhou Youlan Marine Biotechnology Co., Ltd., Guangzhou, China) (140 mg/kg/d). The body weight of each mouse was recorded weekly, and the food intake was monitored twice a week. Four weeks later, the mice were fasted for 12 h and were sacrificed, and then the components of the blood serum from the mice of each group was separated by centrifugation at 1500 r/min for 30 min at 4 °C while the T2DM model were fed a high-fat diet (basic diet, 78.8%; lard, 10%; egg yolk powder, 10%; cholesterol, 1%; and deoxycholate, 0.2%). During the experiment, the T2DM group and COS group mice were intraperitoneally injected with 100 mg/kg streptozotocin (STZ, dissolved in citrate buffer, pH 4.4) in the 2nd week, and T2DM model mice were established, which was deemed successful when the mouse fasting blood glucose (FBG) levels were ≥11.1 mmol·L. Then, the NFD and HFD group mice were intragastrically administered normal saline, and the COS group mice were intragastrically administered with snow crab-derived chitooligosaccharides (Guangzhou Youlan Marine Biotechnology Co., Ltd.) (140 mg/kg/d). The body weight of each mouse was recorded weekly, and the food intake was monitored twice a week. Five weeks later, the mice were fasted for 12 h and were sacrificed, and then, the components of the blood serum from the mice of each group was separated by centrifugation at 1500 r/min for 30 min at 4 °C. Total cholesterol (TC), triglyceride (TG), density lipoprotein cholesterol (LDL-C), and (high-density lipoprotein cholesterol (HDL-C) were measured using assay kits (Biosino Bio-technology and Science Incorporation, Beijing, China) according to the manufacturers’ instructions. Furthermore, the atherogenic index (AI) and the anti-atherogenic index (AAI) were calculated as follows: AI = TC/HDL-C-1, AAI = HDL-C/TC.

In addition, the liver and intestinal tissues of mice were also collected and immediately frozen in liquid nitrogen for further analysis. Total genomic DNA from contents of intestine was extracted and V3 + V4 regions of bacterial 16s rDNA (from 341 to 806) were amplified and sequenced by Illumina Hiseq 2500 PE250 (Genedenovo Inc., Guangzhou, China). All the procedures performed in the animal experiments were in accordance with the ethical standard.

### 4.4. Extraction of Total RNA and RT-qPCR

Total RNA was extracted with TRIzol reagent (Invitrogen, Carlsbad, CA, USA) under RNase-free conditions. Then, Moloney Murine Leukemia Virus (M-MLV) reverse transcriptase (Promega) was used to synthesize cDNA, and each cDNA sample was quantified by real-time qPCR using the Biosystems StepOne™ real-time PCR system and Fast SYBR Green Master Mix (Applied Biosystems, Foster City, CA, USA). PCR primers were designed with National Center for Biotechnology Information (NCBI) online software Primer-BLAST and synthesized by Invitrogen. The sequences of primers used in this study are listed in Table 1.

### 4.5. Western Blotting

Proteins in cell lysates were separated by SDS-PAGE and then transferred to nitrocellulose membranes (Millipore, Burlington, MA, USA). The primary antibodies included 3-hydroxy-3-methylglutaryl-coenzyme (HMGCR) (rabbit antihuman monoclonal antibody; dilution, 1:1000; catalogue No. ab174830; Abcam, Cambridge, MA, USA), SET and MYND domain containing 3(SMYD3) (rabbit antihuman monoclonal antibody; dilution, 1:1000; catalogue No. ab187149; Abcam, Cambridge, MA, USA), and glyceraldehyde-3-phosphate dehydrogenase (GAPDH) (Santa Cruz). Secondary antibodies were IR dye-conjugated donkey antimouse or antirabbit immunoglobulin G (IgG) (LI-COR Biosciences, Lincoln, NE, USA), and proteins on the nitrocellulose membranes were visualized with an Odyssey Infrared Imaging System (Gene Company, Hong Kong, China).

### 4.6. Luciferase Reporter Assay

The pGL3-HMGCR-luciferase reporter plasmids carrying the wild-type HMGCR promoter sequence were constructed with pGL3-basic firefly luciferase using EcoRV and HindIII sites. Then, pGL3-HMGCR-mut-luciferase plasmids were generated with a site-directed mutagenesis kit (TransGen, Beijing, China). HepG2 cells were transfected with pcDNA3.1-SMYD3 or si-SMYD3, as indicated, 24 h before the transfection of the luciferase reporter plasmids. Twenty-four hours after the transfection of luciferase plasmids, the luciferase activity was measured with a luciferase assay system (Promega, Madison, WA, USA) in a Synergy™ 4 Luminometer (BioTek, Winooski, VT, USA). For drug treatment, HepG2 cells were transfected with luciferase reporter plasmids for 24 h, and COS were added 6 h before the measurement of the luciferase activity.

### 4.7. Chromatin Immunoprecipitation (CHIP)

CHIP assays were carried out following a standard protocol. Briefly, HepG2 cells were cross-linked with 1% formaldehyde (Sigma, Steinheim, Germany) for 15 min at room temperature with gentle shaking. Cells were harvested with a cell scraper, and the chromatin was fragmented to an average of 300 bp in length with a sonicator. Diluted whole cell sonicates were incubated with anti-SMYD3 (Abcam, ab187149) or anti-RNAPII antibodies (MMS-126R-200) (Covance, Madison, WI, USA). After the protein A beads (Millipore, 17-295) were bound, washed, and eluted, the CHIP products were purified and measured by real-time qPCR.

### 4.8. Oil Red O Staining

HepG2 cells were grown in 24-well plates, fixed with phosphate buffered paraformaldehyde solution (Roth), washed with PBS, and stained with oil red O solution (stock solutions: 50 mg oil red O powder in 100 mL isopropyl) for 15 min at 37 °C. After washing with PBS, the cells were photographed under a microscope. For quantification, oil red O was dissolved by adding isopropanol (100%) to the samples. Then, 100 μL of the supernatant was added to each 96-well plate and measured at 510 nm using a Fluostar OPTIMA (BMG Labtech, Offenburg, Germany) [44].

### 4.9. Statistical Analysis

All statistical analyses were performed using SPSS 17.0 (SPSS, Chicago, IL, USA) using either a one-sample t-test or one-way ANOVA. All data are presented as the mean ± SD (*n* = 8).

## Figures and Tables

**Figure 1 marinedrugs-18-00069-f001:**
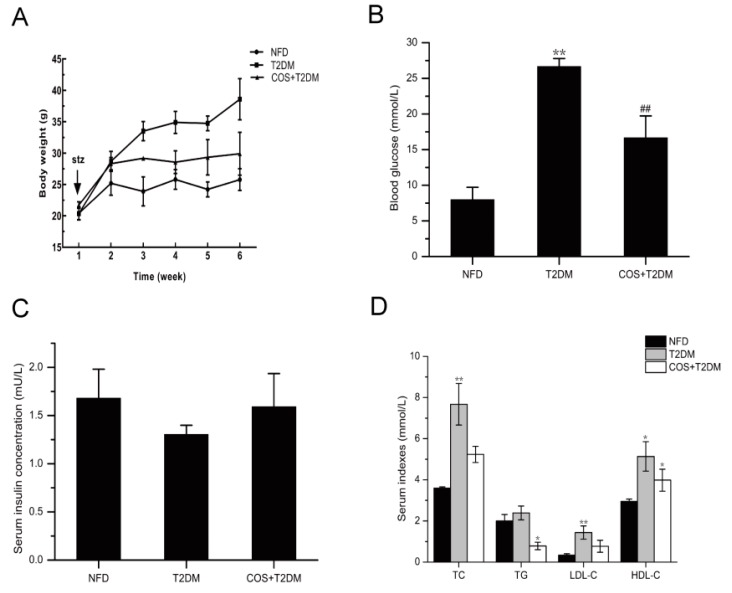
COS ameliorates hyperglycemia and dyslipidemia of T2DM mice. T2DM model was established by feeding mice with high-sugar and high-fat diet in combination with streptozocin (STZ) injection. For COS treatment group, T2DM mice were administrated with 140 mg/g/d for 5 weeks. During the experiment, (**A**) body weight, (**B**) blood glucose levels, (**C**) serum insulin, and (**D**) serum TC, TG, LDL-C, and HDL-C levels were detected. Data are presented as mean ± SD (*n* = 8); * *p* < 0.05 and ** *p* < 0.01, compared with the NFD group; ^#^
*p* < 0.05 and ^##^
*p* < 0.01, compared with the T2DM group.

**Figure 2 marinedrugs-18-00069-f002:**
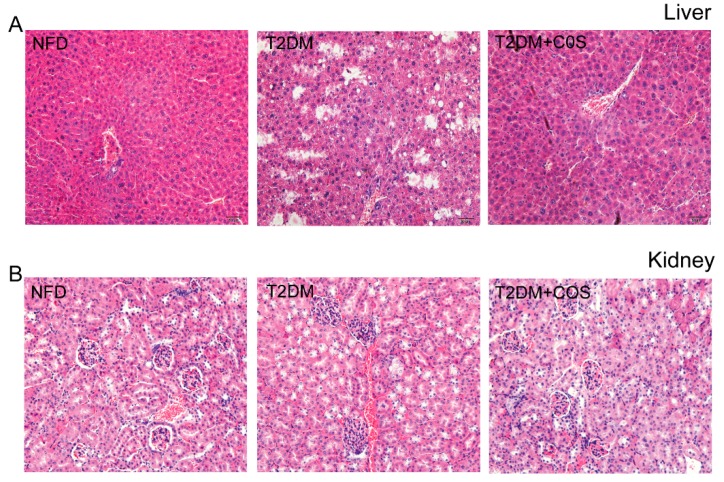
COS protects the liver and renal pathology of type 2 diabetic mice. Pathological detections liver (**A**) and kidney (**B**) were performed by hematoxylin-eosin (HE) staining of histological section.

**Figure 3 marinedrugs-18-00069-f003:**
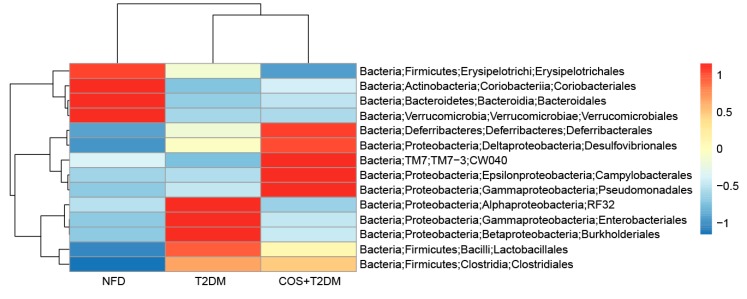
COS reversed T2DM-induced gut microflora dysbiosis. Total genomic DNA from contents of intestine was extracted, variable region3 and variable region4 (V3 + V4) regions of bacterial 16s rDNA were amplified and sequenced by Illumina Hiseq 2500 PE250.

**Figure 4 marinedrugs-18-00069-f004:**
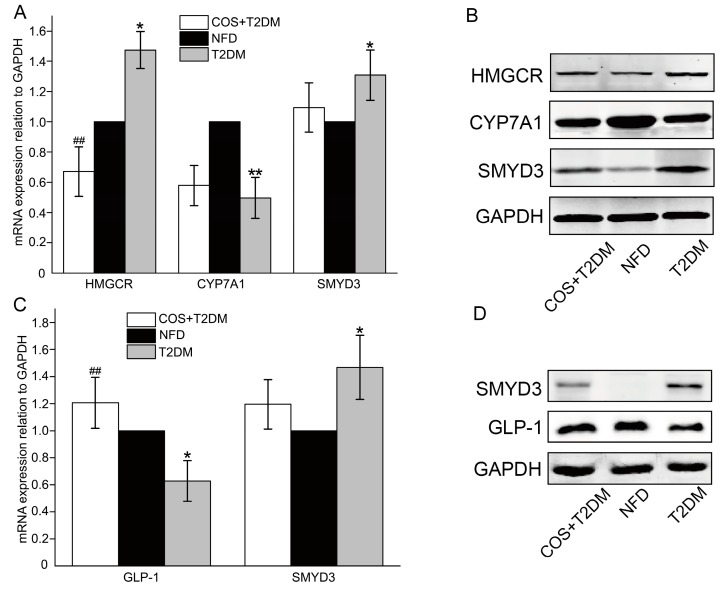
COS regulates lipogenesis-related genes in the liver of T2DM mice. (**A**,**B**) The mRNA and protein levels of the HMGCR, CYP7A1, and SMYD3 in liver were detected by RT-qPCR and western blotting. (**C**,**D**) The mRNA and protein levels of GLP-1 and SMYD3 in intestine were detected by RT-qPCR and western blotting. Data are presented as mean ± SD (*n* = 8); * *p* < 0.05 and ** *p* < 0.01, compared with the NFD group; ^#^
*p* < 0.05 and ^##^
*p* < 0.01, compared with the T2DM group.

**Figure 5 marinedrugs-18-00069-f005:**
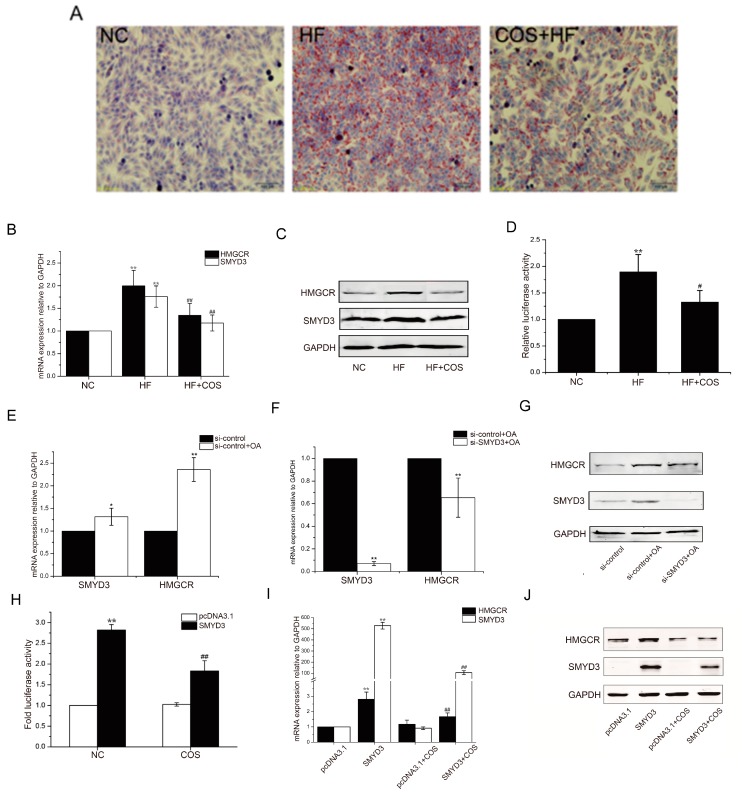
COS inhibits lipogenesis via suppression of SMYD3 and HMGCR in vitro. The high steatosis model of HepG2 liver cells was established by oleic acid induction, and the lipid accumulation was determined by oil red (O) staining (**A**). The mRNA and protein levels of HMGCR and SMYD3 and the transcriptional activity of HMGCR promoter during the oleic acid-induced lipid accumulation were detected by RT-qPCR (**B**), Western blotting (**C**), and luciferase reporter assay (**D**), respectively. Effects of RNA interference (RNAi)-mediated suppression of endogenous SMYD3 on the oleic acid-induced upregulation of HMGCR and SMYD3 were also examined (**E**–**G**). Furthermore, effects of SMYD3 overexpression and COS treatment on the transcriptional activity of HMGCR promoter (**H**), mRNA (**I**), and protein (**J**) levels of SMYD3 and HMGCR were also detected. Data are presented as mean ± SD (*n* = 8); In (**B**,**D**), * *p* < 0.05 and ** *p* < 0.01, compared with control group (NC); ^#^
*p* < 0.05 and ^##^
*p* < 0.01, compared with oleic acid-treated group (HF); In (**E**,**F**), * *p* < 0.05 and ** *p* < 0.01, compared with control siRNA-treated group (si-control or si-control + OA). In (**H**,**I**), * *p* < 0.05 and ** *p* < 0.01, compared with pcDNA 3.1 transfected group (NC), ^#^
*p* < 0.05 and ^##^
*p* < 0.01, compared with SMYD3 transfected group (OA).

**Figure 6 marinedrugs-18-00069-f006:**
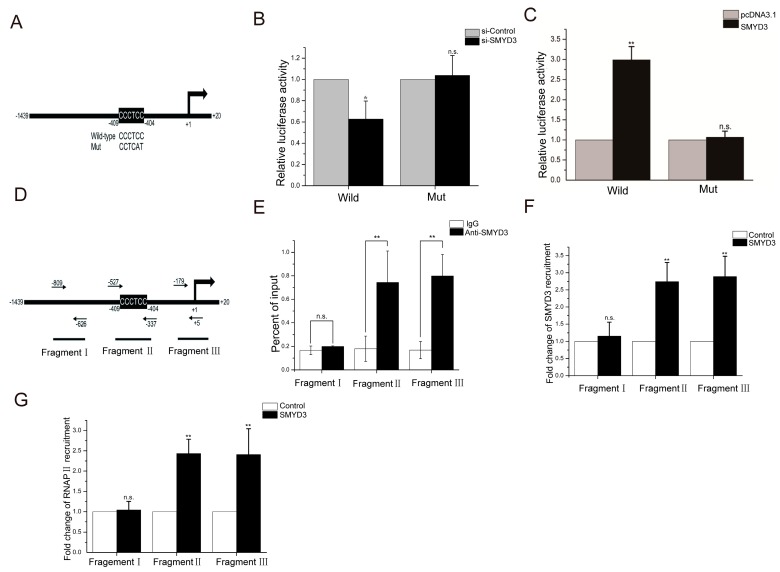
The transactivation of SMYD3 on HMGCR is mainly mediated via the SMYD3 binding site (SBE) site in the promoter. The −1439 to +20 segment of the human HMGCR promoter was cloned and the SMYD3 binding element (SBE) “CCCTCC” was mutated into “CCTCAT” (**A**), and then the effects of siRNA-mediated suppression (**B**) or overexpression (**C**) of SMYD3 on the transcriptional activity of the wild-type or mutated HMGFCR promoter was detected by luciferase reporter assay. CHIP assay were further performed to confirm the association between SMYD3 with the SBE site in HMGCR promoter, quantitative PCR was used to detected 3 different regions of HMGCR promoter after the immunoprecipitation (**D**), the association of endogenous SMYD3 (**E**) and the recruitment of SMYD3 (**F**) or RNA polymerase II (RNAPII) (**G**) to these three sites after SMYD3 overexpression were further analyzed. Data are presented as mean ± SD (*n* = 8); * *p* < 0.05 and ** *p* < 0.01, compared with control (si-Control, pcDNA3.1, IgG or Control); n.s., no significance (*p* > 0.05).

**Table 1 marinedrugs-18-00069-t001:** Primer design for real-time PCR.

Gene	Forward Primer (5′-3′)	Reverse Primer (5′-3′)
GAPDH	CGAGATCCCTCCAAAATCAA	TTCACACCCATGACGAACAT
HMGCR	CTCCTCCTTACTCGATAC	TAGATACACCACGCTCAT
SMYD3	CCCAGTATCTCTTTGCTCAATCAC	TTACGGGTGTTGAAGGT
CYP7A1	CAGAAGCATAGACCCAAGTGAT	TCGGTAGCAGAAGGCATACATC
GLP-1	GATTCAGTCCCAGGCAGCGTAT	CTTTCTTGATCTTGGCGGGTGTT
mHMGCR	TTATGTCTTTAGGCTTGGTC	ACTCAGGGTAATCACTTGC
mGAPDH	ATTCAACGGCACAGTCAAGG	GCAGAAGGGGCGGAGATGA
mSMYD3	CTGCTTTGAGTGTGACT	CTGGTAGATGTTGATGT
Fragment I	CGCTGATTTGGGTCTATG	GTGCGTTCCTTCTGCTCT
Fragment II	ATGGGTAAATCTCGGGAA	GGAGCGTGAGGGAAAACG
Fragment III	GCAGGCCCTAGTGCTGGG	AAGAGAGGATCGTTCGAT

(m: mouse).

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
