# Peer review of "Chitooligosaccharides Modulate Glucose-Lipid Metabolism by Suppressing SMYD3 Pathways and Regulating Gut Microflora"

_marinedrugs, 2020, doi:10.3390/md18010069_

Round 1

Reviewer 1 Report

Comments to the Authors of manuscript number: marinedrugs-670754 entitled Chitooligosaccharides Modulate Glucose-lipid Metabolism by Suppressing SMYD3 Pathways and Regulating Gut Microflora.

In the present paper I reviewed chitooligosaccharides were given intra-gastrically to mice to show how they mediate glucose and lipid metabolism disorders by regulating gut microbiota and SMYD3-mediated signalling pathways.

It seems that the authors knew what they want to show, but there are a few ambiguities.

The description of the study and groups used is not clear and for this reason is difficult to follow the results. It should be corrected.

How many groups are here? The NFD, HFD, COS and T2DM?

L 308 All abbreviation should be explained.

Authors used chitooligosaccharides. However, there is no information what kind. From what animals? How prepared? There is nothing.

L 304 with a lowercase letter – chitooligosaccharides. Figure 2 has no description. Readers do not know what they present. The histopathological description is poor. Pancreas is not presented. There is no figure.

Reviewer 2 Report

Line 42: Reword “initial points”

Line 42: Need Reference after “affect each other”

Line 44: Hyperglycaemia is repeated

Line 46: Clarify what type of diabetes

Line 62: Rephrase “on the other hand..”

Line 75: Start new paragraph.  SMYD3….

Line 92: Hypoglyaemic and hypolipidemic imply that COS will lower these levels below normal. Reword. Indicate that theT2DM is induced by a high fat diet in combination with low-dose streptozotocin.

Line 93: Add dose and time to materials and methods. Was this administered on daily?

Please indicate on Figure 1A when streptozotocin was administered.

Add food consumption data.

Line 98: “additdaemia”?

Line 102: Define TC, TG etc.

Line 105: Define AI, AAI.

Line 113: mean ±SD or SEM? Remove X±s.

Line 116: Referring to Figure 2A. Typically what is seen with high fat feeding is an increase of fat deposition in the liver, indicated by white apota in a H&E stained image. None of that is present in this figure. Unless there is a specific stain for inflammatory cells, there is no evidence of such cells. The liver histology does not indicate any pathology.

Line 121: Similar to he liver, the conclusion make is not justified by the images.

Line 138: Cannot use the word reverse, rather..” COS altered the  T2DM-induced gut microflora dysbiosis.

Line 143: change “while” to ”and”

Line 146-147: Not all indicated genes involved in lipid metabolism, reword sentence.

Figure 4: Change order of groups in 4A to match 4C.

Figure 4A and C, B and D: HMGCR image in 4B is not consistent with results in 4A. Please confirm that this is correct. Add quantitation of Western blots.

Figure 4A and C: Duplicated data for SMYD3, please remove.

Line 158 – What is mean by degeneration of fatty acids?

Line 162 – “aberrantly activated” is not correct here.

Figure 5E: Define OA in legend.

Line 203: Define TSS

Line 204: Not clear what the difference is between Fragment II and III. Does fragment III include the SBE?

Add reference for line 239-240.

 Discussion: Switching between cardiovascular disease, T2D and cancer makes the discussion extremely confusing. 

Line 266: What is meant by normal group.

Materials and methods

Line 282: Be consistent with the company names and locations.

Line 294: Define mouse strain and age

Line 297-298 – What is meant by adaptive feeding?

Line 200 – Define “basic diet”

Line 304 – COS dose? Where COS administered just once?

Author contribution need to be better defined.

Reviewer 3 Report

This work builds on earlier studies of the effects of COS on metabolic disease in mice.  There are several issues with the work that are of major concern:

1) The histology provided in Fig. 2 is not sufficient to support the conclusion that COS protects against renal and liver pathologies.  For example, for liver, serial sections from each mouse would need to be quantitatively analysed for NAFLD using a scoring system such as NAS. 

2)  The cell images in Fig. 5A are reported to be HepG2 cells, which is highly unlikely (https://www.atcc.org/~/media/Attachments/Micrographs/Cell/HB-8065%20Low%20High.ashx). The cells in Fig. 5A appear to be fibroblasts rather than epithelial. This calls into question all corresponding data.

3) Atherogenic and Antiatherogenic Indices are not relevant in wild type mouse strains because cholesterol is predominantly carried in HDL. These indices are only relevant in models which, like humans, carry cholesterol predominantly in LDL.

4)  The mouse strain and diet compositions are not provided.  Basic diet composition must be completely disclosed.

5) Animal ethics approval is not fully disclosed (please provide the name of the animal ethics governing body and protocol approval number).  The T2DM mice in this study exhibited exceedingly high fasting blood glucose (Fig. 1B), and should have been provided exogenous insulin, or at least closely monitored for overt symptoms of hyperglycemia.

6) The Conflict of Interest Statement needs to be modified to clearly indicate that Guangzhou Youlan Marine Biological Technology Co., Ltd is part of the author list as stated on the title page.

Round 2

Reviewer 3 Report

The authors have not responded adequately to some major concerns raised on initial review.

1)   The obvious conflict with Guangzhou Youlan Marine Biological Technology Co., Ltd., as indicated in the author affiliations needs to be clearly declared in the Conflict of Interest Statement.  This is critical for transparency and will not undermine the value of the work.

2) Histopathology images provided in Fig. 2 are not sufficient to conclude COS has the effect of ameliorating the damage of liver and kidney during diabetes development. There is no quantitative data to support this statement.

3) The images of HepG2 cells provided by the authors in rebuttal do not support their data.  The cells shown in Fig. 5A are clearly fibroblasts, and not epithelial in morphology. The fact that they accumulated cytosolic lipid droplets upon exposure to oleate is not unique to HepG2 cells, in fact all cell types will respond in this way to varying degrees.  The authors need to confirm the cell type that was used and report this information.  Again, this is critical for transparency and will not undermine the value of the work.

Round 3

Reviewer 3 Report

The authors have responded appropriately.  The identity of the cell cultures used is still questionable, but the authors have provided enough information for readers to draw their own conclusions.  Please include the paperwork from the supplier indicating the source of the cells as a supplemental file with the final publication.

Author Response

The authors have responded appropriately.  The identity of the cell cultures used is still questionable, but the authors have provided enough information for readers to draw their own conclusions.  Please include the paperwork from the supplier indicating the source of the cells as a supplemental file with the final publication.

Response : Thank you for your comments. The report provided by the cell supplier has been uploaded as supplemental file in the system.